# RankPPGR: automatic diet monitoring through rank learning

Ghady Nasrallah
*Department of Computer Science and Engineering*
*Texas A&M University*
College Station, USA
ghadynasrallah@tamu.edu

Anurag Das
*Department of Computer Science and Engineering*
*Texas A&M University*
College Station, USA
adas@tamu.edu

Sicong Huang
*Department of Computer Science and Engineering*
*Texas A&M University*
College Station, USA
siconghuang@tamu.edu

Bobak J. Mortazavi
*Department of Computer Science and Engineering*
*Texas A&M University*
College Station, USA
bobakm@tamu.edu

Ricardo Gutierrez-Osuna
*Department of Computer Science and Engineering*
*Texas A&M University*
College Station, USA
rgutier@tamu.edu

*Abstract*—Managing diabetes requires careful monitoring of food intake, yet manual logging is burdensome and error-prone. Prior research has shown that the macronutrient composition of a meal (e.g., carbohydrates, protein, fat, and fiber) can be inferred from its postprandial glucose response (PPGR). However, this is a challenging problem given the large inter-individual differences in PPGRs, and the complex interaction between macronutrients in mixed meals. To address these issues, we propose RankPPGR, a rank-learning framework that analyzes within-subject differences in pairwise PPGRs from meals with varying macronutrient composition, and learns a non-linear embedding of meal macronutrients that reflects their joint impact to glycemic responses. We also propose a few-shot regression module that uses outputs from RankPPGR to infer macronutrient composition using a limited number of labeled meals per individual. We evaluate the model on an experimental dataset containing PPGRs to mixed meals from 45 participants. RankPPGR significantly improves both pairwise classification and macronutrient inference performance over a sample-based regression baseline.

*Index Terms*—Diet monitoring, continuous glucose monitors, rank learning.

**Clinical relevance**. Monitoring food intake is an essential component in diabetes management. Current approaches rely on manual entry, which is cumbersome and error-prone. We propose a sensor-based methodology to monitor food intake automatically from continuous glucose monitors. Our approach may also be used to monitor patient adherence to diets in nutritional therapy interventions aimed at adopting low-glycemic load diets.

## I. INTRODUCTION

Diabetes mellitus is a chronic metabolic disease that affects over 500 million adults worldwide [1]. The most common form, type 2 diabetes (T2D), is strongly associated with sedentary behavior, obesity, and poor dietary habits [2]. T2D results from prolonged elevations in glucose levels (hyperglycemia), which arises due to impaired insulin production or reduced sensitivity to insulin. Sustained hyperglycemia can have disastrous long-term health consequences, including cardiovascular diseases (the main cause of death in the developed world), retinopathy, peripheral neuropathy, and nephropathy.

Monitoring and controlling diet is therefore critical to preventing and managing diabetes. However, manual recording of food intake is often time-consuming, error-prone, and difficult to sustain over time [3]. Automated methods using various sensing modalities (e.g., gestures, sound, vision) have been developed to monitor food intake. Among these, continuous glucose monitors (CGMs) represent a promising approach. CGMs capture the increase and subsequent return to baseline of glucose levels when individuals consume meals. The primary contributor to these so-called postprandial glucose responses (PPGRs) is carbohydrate (carb) intake, but other macronutrients (i.e., protein, fat, and fiber) also contribute albeit in the opposite direction, dampening or delaying the glucose spike [4]. This suggests that the shape of a PPGR can be used to infer, in part, the macronutrient content of meals. However, predicting the full macronutrient composition from PPGRs is a complex, many-to-one inverse problem [3]. Moreover, PPGRs are also affected by a variety of factors such as metabolic parameters (e.g., HbA1c, insulin sensitivity), and demographics (e.g., sex, age, ethnicity). As a result, PPGRs exhibit substantial inter-individual variability, even in response to the same meal, which can hinder the generalizability of predictive models across subjects.

To address these two issues, we propose **RankPPGR**, a rank-learning framework that (1) analyzes pairs of PPGRs to meals with different macronutrient contents when consumed by the same individual, and (2) learns a non-linear combination (i.e., an embedding) of macronutrients that can be predicted from PPGRs. By learning to rank meal pairs *within-subject* from raw PPGRs, the model avoids the issue of individual variability and the need for ad-hoc PPGR normalization strategies [5]. Further, by learning which non-linear combination of macronutrients can be extracted from PPGRs, the model avoids the need for feature engineering. We extend this framework with a few-shot learning model that predicts meal compositions with limited data samples per individual.

In a first step, we establish the feasibility of the rank-learning approach using the carb-caloric ratio (CCR) as the dependent variable. The CCR is defined as the ratio of caloric content from carbohydrates to all energetic intake, and therefore of clinical significance in medical nutrition therapy (e.g., adopting low-glycemic load diets) [6]. However, the CCR assumes that non-glycemic macronutrients (i.e., protein, fat and fiber) contribute in the same proportion to PPGRs. Thus, in a second step, we extend the model so it simultaneously learns to rank PPGRs and a non-linear parametric embedding of macronutrients that best reflects information in PPGRs.

We evaluate the performance of the proposed rank-learning

model against an equivalent model that predicts macronutrient compositions in a sample-based fashion, that is, from each individual PPGR. The main contributions of this paper are:

- A rank-learning framework (RankPPGR) that can discriminate between meals with high- and low-glycemic loads.
- A few-shot model that can predict the caloric ratio of carbohydrates to total energetic intake from pairwise ranks of PPGRs
- Validation on an experimental dataset containing nearly 450 postprandial glucose responses to mixed meals.

## II. RELATED WORK

Multiple studies have examined the effect of meal macronutrient amount and composition on PPGRs. The main determinant of postprandial glucose is the amount and type of carbs. Carbs are typically compared by their glycemic index (GI) [7], a measure of how they raise glucose levels compared to a reference food (typically glucose) defined as having a GI=100. Most vegetables have low GIs ($<$55), whereas sugars and starches have high GI ($>$70). However, the GI does not consider the effect of other meal macronutrients. Specifically, adding protein, fat or dietary fiber to a meal reduces and/or slows down the PPGR [8], [9], typically due to gastric emptying or insulin secretion [10]–[12].

Given the broad availability of continuous glucose monitors (CGM) and advances in machine learning (ML), recent work has explored predicting personalized glycemic responses to food. In a seminal study, Zeevi et al. [13] collected CGM data and meal logs from an 800-person cohort. They observed substantial inter-individual variability in PPGRs to identical meals. To address this, they developed an ML model that integrated clinical features, dietary habits, physical activity, and gut microbiome profiles to predict personalized glycemic responses. Their approach was validated in a randomized controlled intervention, where personalized dietary recommendations significantly reduced PPGR excursions. Tily et al. [14] also highlighted the importance of individual-specific features, especially gut microbiome activity, in predicting PPGRs. ML models trained on food composition, anthropometrics, and microbial pathway activity showed that gut microbiome features significantly improved predictions.

These two studies sought to estimate PPGRs to meals given their macronutrient composition (i.e., a *direct* problem). Its *inverse* counterpart, i.e., predicting macronutrient compositions from PPGRs, is far more complex given that meals with different macronutrient content can lead to the same PPGR. In addition to its *inverse* nature, the PPGR-to-macronutrient prediction problem is challenging because of the significant effect that metabolic health parameters (e.g., HbA1c, insulin sensitivity, body mass index, gut microbiota) as well as demographics (e.g., sex, age, ethnicity) play in metabolism. Thus, PPGRs from two individuals cannot be compared without controlling for these health/demographic factors. **The work proposed here address this complex issue by viewing the inverse problem as one of rank learning.**

## III. METHODS

The overall system architecture for RankPPGR is shown in Fig. 1. The model consists of two modules: (1) a Siamese PPGR encoder network, and (2) an aggregator network. During training, the PPGR encoder learns to generate embeddings $z_i, z_j$ from pairs of time-series glucose measurements $g_i(t), g_j(t)$, i.e., PPGRs for meals $m_i, m_j$ such that $z_i > z_j$ if the macro composition of $m_i$ is greater than that of $m_j$. Given a test meal $g_{test}(t)$ with unknown macronutrient composition, the Siamese network generates a vector of probabilities $[p_1, p_2...]$ by comparing $g_{test}(t)$ against sample PPGRs $[g_1(t), g_2(t)...]$ from the subject with known macros. It is this probability vector that the aggregator network uses to estimate the macronutrient composition of the test meal.

To establish proof of concept, we initially train the PPGR encoder and aggregator networks using the carb-caloric-ratio (CCR) as the dependent variable:

$$CCR = \frac{C}{C + P + F + B},\tag{1}$$

where $C, P, F, B$ denotes the *caloric* amount of net carbs, protein, fat and fiber in the meal, respectively. In Section III-C, we replace the CCR (i.e., a "fixed" macronutrient embedding) with one whose parameters can be learned in an end-to-end fashion. The result is an interpretable embedding (a ratio of macronutrients) that best represents how macronutrients contribute to PPGRs.

### A. PPGR encoder

We train the encoder and aggregator networks on the CGMacros dataset publicly available on PhysioNet [15], which contains PPGRs to ten mixed meals from N=45 participants; see section III-F for details. Before training the model, we generate input pairs $g_i(t), g_j(t)$ of PPGRs in a *within-subject* fashion to avoid inter-individual differences. Namely, for each subject in the dataset, we consider all pairs of meals with different CCRs. For each pair, we assign a binary label:

$$y_{ij} = \begin{cases} 1 & \text{if } CCR(g_i(t)) > CCR(g_j(t)), \\ 0 & \text{otherwise.} \end{cases}\tag{2}$$

We train the encoder network following RankNet [16]. Namely, a shared network takes two PPGRs as inputs and predicts which of the two was induced by a meal with a higher CCR. Once trained, the output of the model is a probability score reflecting which of the two PPGRs is more likely to correspond to a meal with higher CCR.

We evaluate for different deep-learning architectures for the shared encoder: multi-layer perceptrons (MLPs), convolutional neural networks (CNNs), gated recurrent units (GRUs) and Transformers. For each model, we vary key hyperparameters (e.g., hidden sizes, number of layers) to produce models with different numbers of trainable parameters. We then evaluate performance as a function of model type and model size (i.e., number of parameters).

Regardless of its architecture, the PPGR encoder produces a latent score $z_i$ for each input $g_i(t)$, from which we compute

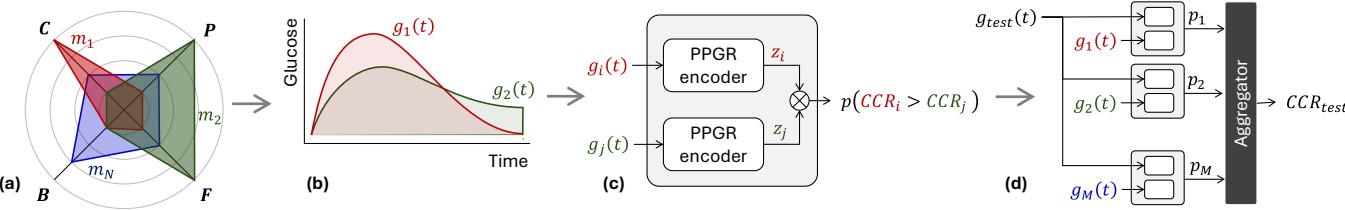

Fig. 1. (a) Polar plot shows the composition of three meals $m_1, m_2, m_3$ with different macro amounts (carbs $C$, protein $P$, fat $F$ and fiber $B$). (b) Postprandial glucose responses to meals $m_1, m_2$. Meal $m_1$ has higher carb-caloric-ratio (CCR) than meal $m_2$, therefore leads to higher PPGR peak with a faster recovery to baseline than meal $m_2$. (c) Architecture of the rank-learning model. The model uses a shared encoder network in a Siamese configuration. (d) Architecture of the aggregator model. The aggregator computes the vector probability that a test PPGR $g_{test}(t)$ has higher CCR than a set of reference PPGRs.

probability that $CCR(g_i(t)) > CCR(g_j(t))$ as $p_{ij} = \sigma(z_i - z_j)$, where $\sigma(x) = (1 + e^{-x})^{-1}$. The model is trained to minimize the binary cross-entropy (BCE) loss between the predicted probabilities and the ground-truth pairwise labels, averaged over all training pairs:

$$\mathcal{L}_{\text{BCE}} = -\left( y_{ij} \log(p_{ij}) + (1 - y_{ij}) \log(1 - p_{ij}) \right), \quad (3)$$

### B. Aggregator network

The aggregator consumes probability estimates from the shared encoder network and converts them into a CCR estimate. Namely, given a test meal $m_{test}$, we compare its PPGR $g_{test}(t)$ against a set of *reference* PPGRs corresponding to a fixed set of known CCR values:

$$\mathcal{R} = \{(g_1(t), CCR_1), \ldots, (g_M(t), CCR_M)\} \quad (4)$$

This results in a vector of probabilities $p_1, p_2...p_M$ for test meal having higher CCR than each of the reference meals[1], which the aggregator uses to estimate the CCR of the unlabeled meal through a regression task. In this fashion, each meal in the training set can be used both as a reference meal and as a test meal[2]. For the aggregator network, we train an MLP to minimize the mean squared error (MSE) between the predicted and ground truth CCR values of the test meals.

### C. Parametric macronutrient encoder

The carb-caloric ratio (CCR) assumes a uniform weighting of macronutrients based on their caloric content. However, the actual magnitude of each macronutrient's contribution to glycemic responses is unknown, even when matched by caloric content. For this reason, the final module in RankPPGR extends the *fixed* CCR in eq. (1) to have learnable coefficients:

$$WCCR(w_P, w_F, w_B) = \frac{C}{C + w_P P + w_F F + w_B B} \quad (5)$$

where $w_P = \text{softplus}(\theta_P)$, $w_F = \text{softplus}(\theta_F)$, and $w_B = \text{softplus}(\theta_B)$, with $\theta_P, \theta_F, \theta_B \in \mathbb{R}$ –see Fig.. 2 This allows

---

[1]In our experiments, we used a fixed reference set of 5 meals, each corresponding to a distinct CCR value. This setup can be generalized to any number of reference meals, provided sufficient PPGR data is available across subjects for each meal to support few-shot comparisons.

[2]Pairs of test/reference PPGRs are taken from the same subject whenever possible. Whenever a reference PPGR with a specific CCR is missing, we replace it with that from another subject with the closest HbA1c. Further, when multiple PPGRs from the same subject are available for a given reference meal, we average the probabilities across those comparisons.

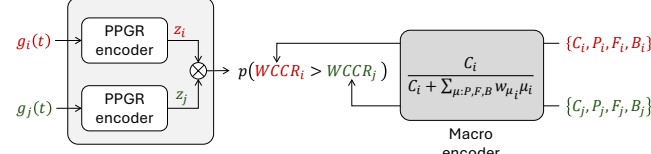

Fig. 2. Learning a parameterized (CCR-based) macronutrient embedding jointly with the PPGR encoder.

the model to find a macronutrient representation that can be best predicted from PPGRs, within the constraints of a CCR calculation that is both clinically interpretable and physiologically plausible. To ensure non-negativity and smoothness, we define each coefficient $w_P, w_F, w_B$ as the softplus of an unconstrained parameter $\theta_P, \theta_F, \theta_B$, respectively, where $\text{softplus}(x) = \log(1 + e^x)$. Parameters $\theta_P, \theta_F, \theta_B$ are initialized such that the corresponding coefficients $w_P, w_F, w_B$ are equal to 1.

We learn the WCCR coefficients following the same pairwise learning framework described earlier. Namely, for each subject, we create pairs of PPGRs from meals not with different CCRs but different macronutrient composition, i.e., $C, B, P, F$. The label for each pair is determined based on the $WCCR$ ranking induced by the current coefficient values:

$$y_{ij} = \begin{cases} 1 & \text{if } WCCR_i > WCCR_j \\ 0 & \text{otherwise} \end{cases} \quad (6)$$

Note that this binary label is not differentiable with respect to the coefficients. For this reason, we apply a sigmoid function over the difference in WCCR values, as:

$$\tilde{y}_{ij} = \sigma\left(\alpha \cdot (WCCR_i - WCCR_j)\right) \quad (7)$$

where $\sigma(x) = (1 + e^{-x})^{-1}$, and $\alpha$ is a tunable parameter that controls the slope of the sigmoid. We train the PPGR encoder and macronutrient encoder jointly using BCE via gradient descent. After the PPGR and macronutrient encoders are trained, we train the aggregator model to predict continuous WCCR values based on pairwise probabilities computed with the optimized WCCR formula.

### D. Validation strategy

We validate the RankPPGR models through cross-validation *across* subjects, as illustrated in Fig. 3. First, we sort subjects in the CGMacros dataset according to HbA1c. Then,

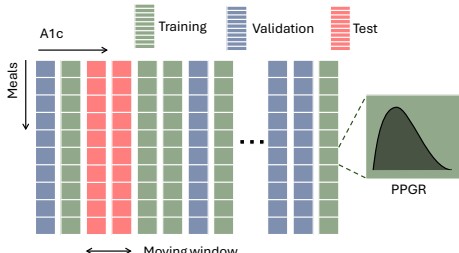

Fig. 3. Validation procedure. We sort subjects by A1c, then select two adjacent subjects (red) for testing using a moving window. From the remaining N-2 subjects, 4 are randomly selected as validation samples (blue) for hyperparameter tuning. The remaining subjects (green) are used for training

we apply a moving window to select two adjacent subjects as held-out samples for testing. This setup enables subject-level generalization and supports the substitution of reference PPGRs across held-out subjects, as described in the aggregator model section, in cases where a given subject lacks a meal corresponding to a specific reference CCR or WCCR.

Within each outer fold, we perform hyperparameter tuning on 4 randomly held out validation subjects from the training set. For the MLP models, we conduct a grid search over hidden layer sizes $(32, 16)$ and $(64, 32)$; learning rates $\{10^{-2}, 10^{-3}\}$; batch sizes $\{16, 32\}$; and weight decay values $\{0, 10^{-4}\}$. For WCCR tuning, we tune the smoothing parameter $\alpha \in \{3, 5, 7\}$.

We evaluate model performance using regression and classification metrics. For regression purposes, we report three metrics: Pearson correlation, the coefficient of determination (R²), and Relative Root Mean Squared Error (RRMSE), defined as:

$$\text{RRMSE} = \sqrt{\frac{1}{N} \sum_{i=1}^{N} \left( \frac{y_i - \hat{y}_i}{y_i} \right)^2} \tag{8}$$

where $y_i$ is the true value, $\hat{y}_i$ is the predicted value, and $N$ is the number of test samples. Thus, the RRMSE represents the percent error relative to the ground-truth value, which is more meaningful when comparing predictions for meals with different CCR or WCCR values.

For classification evaluation, we report accuracy and F1 score on the Siamese encoder, using all possible meal pairs within each test subject. For each pair, we evaluate whether the ordering of the predicted CCR or WCCR values matches the ground-truth ordering. F1 scores are computed based on the proportion of correctly ordered pairs.

### E. Baseline model for comparison

For evaluation purposes, we used a baseline model with the same configuration as the PPGR encoder, except the baseline model is trained to predict CCRs directly from individual PPGRs. When evaluating the baseline model in a classification setting (i.e., which of two PPGRs in a pair has higher CCR?), we follow the same pairwise evaluation strategy used for RankPPGR: after training, we consider all meal pairs within each test subject that differ in CCR. For each pair, we assess whether the predicted CCRs preserve the correct ordering. F1 scores are computed based on the proportion of correctly ordered pairs. In addition to classification metrics, we also

compare the regression performance of the baseline and the RankPPGR aggregator network.

### F. Dataset description

We evaluated RankPPGR and the baseline model on the CGMacros dataset publicly available on PhysioNet [15]. The dataset contains 10 days of CGM recordings for 45 subjects (age: 18-69, T2D: 14/45; pre-diabetes: 16/45; $4.6 \leq HbA1c \leq 8.5$). As part of the screening process, a number of variables were measured for each subject, including body mass index (BMI), glycated hemoglobin (HbA1c), fasting glucose, fasting insulin, triglycerides, cholesterol levels and demographics (age, gender, and race). Subjects wore an Abbott FreeStyle Libre Pro CGM (15-min sampling period) and a Dexcom G6 Pro CGM (5-min) on their upper arm and abdomen, respectively. Both CGMs were blinded to prevent glucose readings from influencing participants. Each subject recorded their meals for 10 days, including breakfast, lunch and dinner. Breakfasts consisted of protein shakes with varying amounts of carbs, protein, fat, and fiber. Lunches were ordered from a local, fast-casual restaurant chain (Chipotle Mexican Grill). The breakfast and lunch meals were designed to cover a range of macronutrient contents. For dinners, participants ate foods of their own choice. To minimize interferences in glucose responses from prior meals, participants were instructed to eat lunch at least 3 hours after breakfast, with only water or coffee (without sugar) in between, and dinner at least 3 hours after lunch. In this study, we only analyze PPGRs from the Abbott CGM for the breakfast meals, whose macronutrient composition are shown in Table I.

## IV. RESULTS

### A. Encoder architecture

First, we examined how the choice of model architecture and size affects the performance of the PPGR encoder on the ranking task, i.e., identify which of two PPGRs is from the meal with higher CCR. Results are summarized in Fig. 4 in terms of the average F1 score across test folds as a function of model size. For each model architecture, we repeated the procedure five times and report the average results. Despite having a simple architecture, MLPs achieve the highest F1 score across the range of model sizes. The CNN and Transformer architectures perform comparably to the MLP model

TABLE I
MACRONUTRIENT CONTENT OF BREAKFAST SHAKES, CODED AS HIGH (H) OR LOW (L) AMOUNTS OF NET CARBS, PROTEINS, FAT, AND FIBER [17].

| Day | Meal | Carbs (g) | Prot (g) | Fat (g) | Fiber (g) | CCR |
|---|---|---|---|---|---|---|
| 4 | HHHH | 66 | 66 | 42 | 07 | 0.287 |
| 5 | LLLL1 | 24 | 22 | 11 | 00 | 0.345 |
| 9 | LLLL2 | 24 | 22 | 11 | 00 | 0.345 |
| 10 | HLHH | 66 | 22 | 42 | 07 | 0.355 |
| 3 | HLHL1 | 66 | 22 | 42 | 00 | 0.362 |
| 8 | HLHL2 | 66 | 22 | 42 | 00 | 0.362 |
| 2 | HHLL1 | 66 | 66 | 11 | 00 | 0.424 |
| 7 | HHLL2 | 66 | 66 | 11 | 00 | 0.424 |
| 1 | HLLL1 | 66 | 22 | 11 | 00 | 0.591 |
| 6 | HLLL2 | 66 | 22 | 11 | 00 | 0.591 |

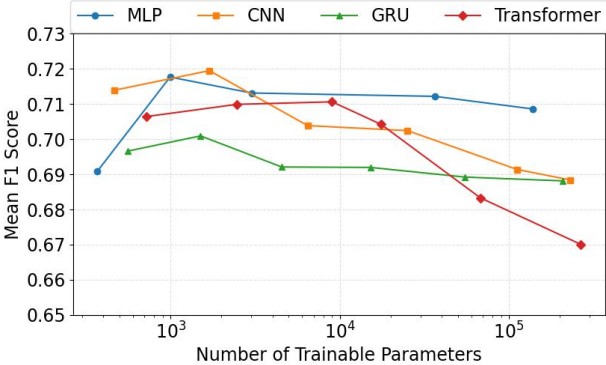

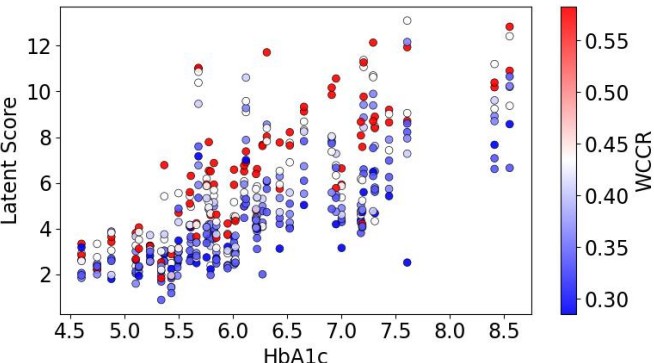

Fig. 4. Average F1 score across test folds for different encoder architectures vs. number of trainable parameters.

Fig. 5. Latent RankPPGR scores plotted against subject HbA1c, with points color-coded by WCCR.

for modest model sizes but their performance degrades as parameters increase, suggesting potential overfitting, with the GRU architecture showing the weakest performance of all models. These results indicate that the more complex model architectures offer limited benefits for the relatively short PPGR sequences used here and dataset size. For this reason, we use an MLP for the remaining sections of this manuscript.

### B. Rank- vs. sample-based classification

In a second step, we compared RankPPGR against the baseline model on the same binary classification task. As shown in Table II, RankPPGR outperforms the baseline model (F1 score of $0.72 \pm 0.10$ vs. $0.65 \pm 0.08$; paired t-test: $p < 0.0001$), averaged across test folds using the leave-2-subjects-out procedure described earlier. Because the main difference between the two models is in the number of inputs (i.e., pairs of PPGRs for our model, individual PPGRs for the baseline), these results indicate that discrimination of low- vs. high-glycemic loads is better approached as a rank-learning task in a *within-subject* fashion to avoid the confounding effect of inter-individual differences and metabolic health across participants. While binary classification accuracies of 0.72/0.65 may appear to be modest, notice that differences in CCRs across meals are in some cases in the second decimal digit –see Table I.

### C. Regression performance (aggregator model)

In a third step, we compare the aggregator and baseline models in terms of their ability to estimate CCRs. We report the *average* $\pm$ *stdev* across folds using a leave-2-subjects-out procedure. Table II summarizes the results for Pearson correlation ($r$), coefficient of determination ($R^2$), and relative RMSE (RRMSE). The aggregator model outperforms the baseline model on all measures. A two-tailed paired *t*-test confirms that the improvements are statistically significant for the three measures ($p < 0.05$). In particular, the baseline model can only explain 6% of the variance in the data, whereas the aggregator explains 4 times as much (25% of the variance). As we will see in Section IV-D , this is likely due to the large inter-individual differences in postprandial glucose across participants, which in our study is amplified by the broad range of metabolic health

parameters in the CGMacros dataset, with HbA1c between 4.5 and 8.5. As a reference, HbA1c<5.7% is considered normal (healthy adults), 5.7%<HbA1c<6.4% is considered pre-diabetes, and HbA1c>6.4% is considered type 2 diabetes.

### D. Structure of latent scores

To interpret the RankPPGR model, we trained the model on the full dataset and analyzed the scalar latent scores produced for each PPGR. Fig. 5 shows the latent score for each PPGR, color-coded by WCCR, plotted against the subjects' HbA1c. Two trends emerge: an increase in latent scores with HbA1c (left-to-right) with corresponding increase by glycemic load (bottom-to-top). This pattern indicates that RankPPGR learns a representation that captures both subject-level metabolic state and meal glycemic load. This latent structure helps explain why our aggregator model outperforms the baseline model. Whereas the baseline model attempts to learn a direct mapping from PPGRs to glycemic load, without accounting for inter-individual variability in glycemic responses and metabolic health status, the aggregator model compares PPGRs across meals from individuals with similar HbA1c values. In the latent space, this corresponds to comparisons along the vertical axis of Fig. 5. Direct comparison of PPGRs from meals with different CCRs across individuals with different HbA1c, as the baseline model attempts to do, is problematic because the two variables (HbA1c and CCR) are confounded when examining postprandial glucose. Instead, RankPPGR compares PPGRs within phenotype-matched groups, thus focusing on information along the vertical axis in Fig. 5, ignoring the confounding effect of HbA1c along the horizontal axis.

### E. Optimizing macronutrient encodings

Following the same leave-2-subjects-out procedure, we trained the PPGR encoder and the macronutrient embedding in eq. (5) *jointly*. The distribution of learned coefficients for

TABLE II
COMPARISON OF RANKPPGR/AGGREGATOR VS. BASELINE ON BINARY CLASSIFICATION AND REGRESSION OF CCR. ($\mu \pm \sigma$) ACROSS TEST SUBJECT FOLDS. $^*p < 0.05$, $^{***}p < 0.001$

| Model | Classification | Regression | | | |
|---|---|---|---|---|---|
| | F1*** (↑) | RRMSE* (↓) | MAE* (↓) | $r$*** (↑) | $R^2$*** (↑) |
| Baseline | $0.65 \pm 0.08$ | $0.23 \pm 0.04$ | $0.08 \pm 0.008$ | $0.39 \pm 0.24$ | $0.06 \pm 0.26$ |
| Aggregator | $\mathbf{0.72 \pm 0.10}$ | $\mathbf{0.20 \pm 0.03}$ | $\mathbf{0.07 \pm 0.006}$ | $\mathbf{0.53 \pm 0.17}$ | $\mathbf{0.25 \pm 0.19}$ |

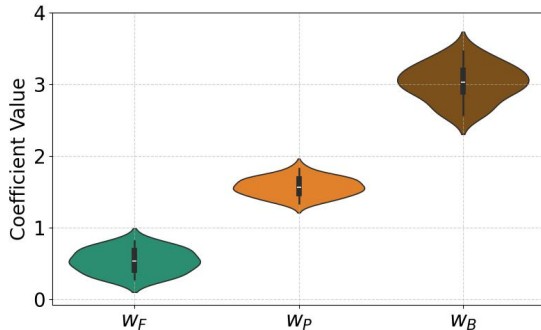

Fig. 6. Distribution of learned macronutrient coefficients ($w_F$, $w_P$, $w_B$). The effect of fat, protein and fiber is $0.5\times$, $1.5\times$ and $3\times$ of what the standard CCR calculation assumes, respectively.

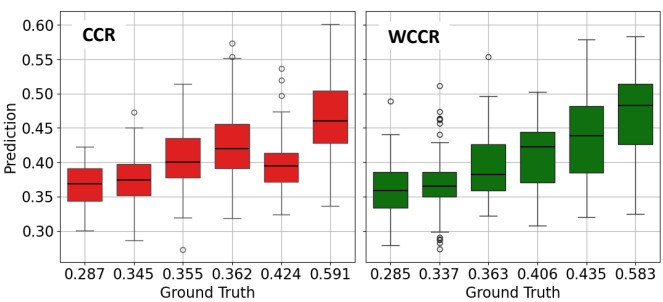

Fig. 7. Box plots of predicted CCR/WCCR grouped by ground-truth. WCCRs have a monotonic relationship between predictions and ground truth, indicating that WCCRs are a better representation of the information embedded in PPGRs than the standard CCR calculation.

TABLE III
COMPARISON OF CCR VS. WCCR ON BINARY CLASSIFICATION AND REGRESSION OF RANKPPGR/AGGREGATOR

| Embedding | Classification | Regression | | |
| --- | --- | --- | --- | --- |
| | F1** (↑) | RRMSE** (↓) | MAE* (↓) | $r^{***}$ (↑) | $R^{2**}$ (↑) |
| CCR | 0.72 ± 0.10 | 0.20 ± 0.04 | 0.07 ± 0.006 | 0.53 ± 0.17 | 0.25 ± 0.19 |
| WCCR | **0.77** ± 0.08 | **0.18** ± 0.03 | **0.06** ± 0.009 | **0.61** ± 0.16 | **0.35** ± 0.20 |

protein ($w_P$), fat ($w_F$) and fiber ($w_B$) across cross-validation folds is shown in Fig. 6. The model consistently assigns the highest weight to fiber ($w_B = 3.06$), followed by protein ($w_P = 1.56$), with fat receiving the lowest weight on average ($w_F = 0.54$). These results are significant as they indicate that the caloric content per gram of macronutrients is not the main determinant of postprandial glucose responses.

Given that WCCR has a stronger association with postprandial glucose responses than the conventional (unweighted) CCR, we reassessed the performance of RankPPGR when using either of the two macronutrient embeddings (WCCR vs. CCR). For this purpose, we evaluated classification performance on pairwise comparisons between meals to determine whether the model correctly identifies the meal with the higher CCR/WCCR. Results are shown in Table III in terms of F1 scores across all test folds. RankPPGR achieves higher F1 scores with the tuned WCCR embedding than with the fixed CCR embedding. A two-tailed paired *t*-test confirms that the performance differences are statistically significant ($p < 0.01$). This indicates that learning optimal macronutrient weights leads to a CCR formulation that is more closely aligned with the actual glycemic impact of the meals.

Likewise, we reassessed the performance of the aggregator model when using fixed CCR vs. tuned WCCR embeddings. Results are summarized in Table III across test folds. The aggregator model achieves lower RRMSE and higher correlation ($r$) and coefficient of determination ($R^2$) when using the tuned WCCR embedding, as compared to the fixed version. A two-tailed paired *t*-test confirms that all differences between the fixed and tuned models are statistically significant ($p < 0.01$).

To corroborate the results in Table III, we also visualized the alignment between predicted and actual CCR/WCCR values. Results are shown in Fig. 7 with box plots of predicted CCR/WCCR grouped by ground-truth CCR/WCCR. With the standard CCR calculation, the model estimates that the fifth meal in group (CCR=0.424) has lower CCR than the preceding meal in the sequence (CCR=0.362). Instead, the WCCR formulation results in a monotonic relationship between predictions and ground truth, indicating that the WCCR macronutrient embedding is a more accurate reflection of how macronutrients contribute to postprandial glucose responses.

### F. Ablation on reference meal availability

To evaluate the practicality of RankPPGR in real-world scenarios where only a limited number of reference meals are available from phenotypically matched subjects, we conducted an ablation study varying the number of reference meals from 1 to 5 (the original setup). This analysis focuses on WCCR prediction using the aggregator model. As shown in Table IV, model performance degrades only modestly as the number of reference meals decreases. Even with just 2 or 3 reference meals, the model retains most of its predictive power. These results suggest that RankPPGR remains robust and practical for use in low-data scenarios such as cold-start deployments.

### V. DISCUSSION

Monitoring dietary intake is a major component of diabetes interventions and nutrition therapies that promote the adoption of low-glycemic load diets [6]. Conventional approaches to monitoring diet require manual entry (food journals), which is not only error-prone but places a high burden on patients. Several technological solutions have been developed to assist in automatic diet monitoring (e.g., accelerometers, gyroscopes, microphones and cameras), but these modalities generally only detect moments of food intake and not the compositions of those foods. Instead, continuous glucose monitors (CGMs) measure the effect of food choices on postprandial glucose, which is a more direct measurement of food intake. The main

TABLE IV
ABLATION STUDY ON THE NUMBER OF REFERENCE MEALS USED FOR WCCR PREDICTION. ($\mu \pm \sigma$) ACROSS TEST FOLDS.

| # Ref. Meals | RRMSE (↓) | $r$ (↑) | $R^2$ (↑) |
| --- | --- | --- | --- |
| 1 | 0.20 ± 0.03 | 0.54 ± 0.14 | 0.21 ± 0.21 |
| 2 | 0.20 ± 0.03 | 0.57 ± 0.16 | 0.26 ± 0.21 |
| 3 | 0.19 ± 0.03 | 0.57 ± 0.16 | 0.27 ± 0.24 |
| 4 | 0.19 ± 0.03 | 0.59 ± 0.18 | 0.31 ± 0.23 |
| 5 (original) | **0.18** ± 0.03 | **0.61** ± 0.16 | **0.35** ± 0.20 |

challenge in CGM-based diet monitoring is that an identical meal can lead to very different postprandial glucose depending on the individual's metabolic health status. Our proposed solution addresses this challenge by treating the problem as one of rank learning. By comparing PPGRs within-subject or between-subjects with similar metabolic health status (HbA1c in our study), RankPPGR avoids the confounding effect of individual differences and glycemic load of meals. We validated RankPPGR on an experimental dataset of PPGRs with different glycemic loads (i.e., carb-caloric-ratio) measured in free-living conditions from patients with a broad range of metabolic health status. RankPPGR consistently outperforms a baseline model (with the same DL architecture but which processes individual PPGRs) using binary classification (pairwise ranking) and regression (prediction of CCRs) metrics.

The relationship between macronutrients in mixed meals and postprandial glucose is complex and non-linear. A conventional measure of glycemic load is the ratio of calories from carbohydrates relative to total calories in the meal (CCR). We find that CCR is not the optimal "embedding" of macronutrients in a meal as it overestimates the effect of fats relative to protein and fiber in a meal. This finding is important for interventions aimed at reducing elevated glucose levels after a meal (post-prandial hyperglycemia) [18], because they suggest protein and fiber are significantly more effective than fat at reducing postprandial glucose, let alone healthier alternatives. According to our results on the CGMacros dataset, protein is nearly 3 times more effective than fat ($w_P/w_F = 1.56/0.54 = 2.88$) and fiber nearly 6 times more effective than fat ($w_B/w_F = 1.56/0.54 = 5.67$). These learned WCCR weights can help clinicians design more effective dietary interventions in nutritional therapy, as they better reflect the physiological impact on PPGR, particularly in distinguishing levels of glycemic load.

## VI. Limitations and future work

While our few-shot approach only requires a limited amount of data, it assumes that reference meals with different macronutrient combinations are available for each subject or from subjects with similar metabolism or health status. This limits the model's applicability to scenarios where phenotype-matched reference data is available.

Future work will extend this framework to zero-shot scenarios by integrating phenotype variables directly into the RankPPGR model. For instance, incorporating subject-level features such as A1c, age, or sex via a feature-wise linear modulation (FiLM) layer could allow the model to account for inter-individual variability explicitly. This would enable cross-subject comparisons without requiring phenotype-matched reference CGM data, thereby improving scalability and generalizability in real-world applications.

## VII. Conclusion

Rank learning is an effective strategy to address inter-individual differences in food metabolism. Given a small number of reference meals from a given subject, or phenotype-matched subjects, our model can be used to discriminate meals with low- and high-glycemic loads. Our results indicate that adding protein and fiber to a meal is far more effective at reducing post-prandial hyperglycemia than fats.

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
