# OpenReview forum: "RankPPGR: automatic diet monitoring through rank learning"
_IEEE.org/EMBS/BHI/2025/Conference — BHI 2025_

### Official Review · Reviewer_EFgG · 2025-07-17
**Review of "RankPPGR: automatic diet monitoring through rank learning"**

**Confidence:** 4
**Clarity Of Writing:** good
**Clinical Significance:** good
**Methodological Novelty:** great
**Overall Rating:** 8
**Final Rating:** 8

**Experiments And Results:**

great

**Questions For The Authors:**

•  While the technical relevance of the proposed approach is clearly demonstrated, its clinical significance remains less well contextualised. For instance, in Table 2, RankPPGR achieves a lower RRMSE than the baseline. However, it would be helpful to interpret the practical meaning of this difference. Specifically, how clinically meaningful is an RRMSE of 0.20 in the context of macronutrient estimation? Is there a known threshold for minimum clinically important error in the estimation of carb-caloric ratios in meals?

**Strengths:**

The study addresses the critical and globally relevant challenge of dietary monitoring, with direct implications for metabolic disorders such as type 2 diabetes.  The proposed framework is methodologically sound, and the paper is generally well written, despite minor attention errors.  The approach has translational potential beyond diabetes care, with possible applications in broader nutritional and dietetic monitoring domains.

**Summary Of The Paper:**

The paper introduces the clinical importance of glycaemic control in diabetes, highlighting the limitations of traditional manual monitoring methods. Continuous glucose monitors (CGMs) are presented as a promising alternative for tracking postprandial glucose response (PPGR), which reflects the body's glycaemic reaction to meal composition. Given that PPGR is primarily driven by carbohydrate intake and modulated by other macronutrients, it is hypothesised that macronutrient composition could be inferred from PPGR curves. However, this inference is complicated by individual-specific noise factors such as age and sex.
To address this challenge, the study proposes a novel framework named RankPPGR, which integrates a Siamese PPGR encoder, an aggregator network, and a few-shot learning algorithm to model non-linear associations between raw PPGR-meal pairs and macronutrient content at participant level.
Evaluation on an experimental dataset demonstrates that the model improves both the discrimination between meals with high versus low glycaemic loads and the estimation of macronutrient content based on individual PPGR profiles.

**Weaknesses:**

•  In the subsection "Clinical Relevance", the acronym “CGM” appears without prior definition. It should be written out in full at first use to aid clarity for non-specialist readers.
•  In the methods section, specifically subsection “B. Aggregator Network", the equation currently lacks a numerical label and should be numbered as (4). Subsequent equations should be renumbered accordingly, in line with IEEE BHI formatting requirements. The same issue is present in subsection “D. Validation Strategy”.
•  Across the methods and results sections, several formatting inconsistencies are evident. For example, Figure 3 is not referenced in the main text, and Table 1 appears before it is first mentioned and exceeds the column width. Tables 3 and Figures 4, 5, and 7 are also placed before their textual citations. IEEE BHI guidelines on figure and table placement should be consulted and followed.
• In Table 2, the symbol used for Pearson's correlation coefficient is denoted as ρ (rho), which is typically reserved for Spearman’s rank correlation. The more standard notation “r” should be used instead.

---

### Official Review · Reviewer_B34J · 2025-07-17
**Novel rank-learning approach for diet monitoring shows promise but limited by dataset scope and modest performance gains.**

**Confidence:** 4
**Clarity Of Writing:** good
**Clinical Significance:** great
**Methodological Novelty:** great
**Overall Rating:** 6
**Final Rating:** 7

**Experiments And Results:**

good

**Questions For The Authors:**

1. How sensitive is the model to inaccuracies or variability in the self-reported macronutrient composition of meals? If the model's performance degrades substantially, its utility in real-world scenarios with noisy labels may be limited.
2. How does RankPPGR perform on non-liquid meals (e.g., lunches or dinners)? Extending the analysis beyond breakfast would improve evidence of robustness.
3. Does the model benefit from incorporating additional features like age or BMI directly in the encoder or aggregator modules? If so, this could help in developing a scalable zero-shot variant.

**Strengths:**

1. Introduces a novel framing of the inverse problem (PPGR to macronutrients) as a rank-learning task, mitigating inter-subject variability.
2. Employs a clinically interpretable and physiologically meaningful embedding (WCCR) that is learned directly from data.
3. Compares multiple encoder architectures and includes a statistically validated performance gain over a strong baseline.

**Summary Of The Paper:**

The paper presents RankPPGR, a rank-learning framework designed to estimate macronutrient composition of meals from postprandial glucose responses (PPGRs) collected through continuous glucose monitors (CGMs). The model compares pairs of glucose responses from the same individual to learn a latent ranking based on carb-caloric ratio (CCR) or a learned weighted CCR (WCCR). An aggregator module then uses these pairwise probabilities to infer the macronutrient profile of a new meal. The approach was evaluated on the CGMacros dataset, which contains labeled glucose responses from 45 participants with varying metabolic health. The model was tested using cross-validation across subjects and showed improved F1 scores and regression metrics compared to a baseline that predicts CCR directly from individual PPGRs. Learned macronutrient weights indicated that fiber and protein have a greater impact on glycemic responses than fat.

**Weaknesses:**

1. External validation beyond the CGMacros dataset is not provided, which raises concerns about generalizability.
2. A dataset size of 45 subjects may not capture wider metabolic diversity.

---

### Official Review · Reviewer_ZNup · 2025-07-18
**Good Paper Filled with Many  Experiments**

**Confidence:** 3
**Clarity Of Writing:** great
**Clinical Significance:** good
**Methodological Novelty:** great
**Overall Rating:** 7
**Final Rating:** 7

**Experiments And Results:**

excellent

**Questions For The Authors:**

- Is there any way that this model predicts the total amount of food/calories consumed based on the PPGR? Or is this paper focused on predicting a specific glycemic metric (e.g. WCCR) of the meal?

- Is there a justification for the sole ranking of participants to be A1c? Given the list of related work with other demographic and gut microbiome features shown to cause variability in PPGR, should any of these or a combination of these be used to pick the closest second subject for the leave-2-subjects-out protocol?

**Strengths:**

- The paper has many experiments and comparisons, thoroughly exploring the comparisons of the ranked model vs the non-ranked baseline.

- The optimized macronutrient encoding results improves the robustness of the results by demonstrating that selecting the correct target can help the model to more easily rank the meals.

**Summary Of The Paper:**

This paper re-frames dietary monitoring based on postpradial glucose responses as a Ranked Leaning problem rather than as a direct prediction. The model is trained on a dataset of Continuous Glucose Monitors worn by 45 participants who each eat 30 meals with known nutritional content. The rank learning model takes a pair of PPGR sequences and outputs the probabilities of each input having a higher carb-caloric ratio (CCR), and a second aggregator is used to rank input meals against a set of 5 reference meals of the same participant (or a participant with very similar HbA1c). A Multilayer Perceptron network was used the ranked encoder, and a weighted CCR was investigated to see if the ratios of macronutrients could be tuned to improve ranking performance. The paper presents modest results due to the difficulty of the problem, but consistently demonstrates how using the ranked model outperforms the baseline direct prediction based model.

**Weaknesses:**

No major weaknesses with methods of this paper, but a few questions listed below about some of the potential applications of the research are asked in the “Questions to the Authors”. These could potentially be addressed in the paper.

---

### Official Review · Reviewer_1FjM · 2025-07-18
**RankPPGR: automatic diet monitoring through rank learning**

**Confidence:** 5
**Clarity Of Writing:** fair
**Clinical Significance:** great
**Methodological Novelty:** good
**Overall Rating:** 3

**Experiments And Results:**

fair

**Questions For The Authors:**

- One of my main concerns is that the experimental validation was performed exclusively on breakfast shakes. Could you provide a rationale for excluding the lunch and dinner data? Demonstrating even baseline performance on this more varied, real-world data would substantially strengthen the paper's argument for "automatic diet monitoring."

- Another main concern is the reference meal substitution strategy described in footnote 2. Could you please quantify what percentage of the comparisons made by the aggregator network involved a "borrowed" PPGR from another subject? For instance, how do the regression metrics in Table II change if you only use subjects for testing who have a complete set of reference meals, thus requiring no substitution? This is crucial for me to properly evaluate the few-shot learning claims of the work.

- The architecture of the learnable WCCR in Figure 1(e) is a bit unclear. It depicts a "Macro encoder" block that outputs the weights, but these weights are parameters of the WCCR formula used in the loss function, not a direct output. Could you clarify or revise this diagram to better reflect how the macronutrient composition and the learnable weights are integrated during training?

- While normalizing on HbA1c is a good start, would it make sense to normalize based on one or two other factors as well? e.g. BMI, age, and biological sex are also contributors to how one metabolizes

**Strengths:**

- The core idea of using a rank-learning paradigm to handle inter-subject variability in PPGRs is well-motivated. By focusing on within-subject pairwise comparisons, the model aims to directly tackles the primary source of variance, providing personalized predictions.

- The extension from a fixed CCR to a learnable, weighted CCR (WCCR) is a neat idea. It allows the model to learn a data-driven, physiologically-interpretable embedding of macronutrients. The findings, e.g. Fig 6, show that contribution of other macronutrients such as fiber and protein have a much larger dampening effect per calorie than accounted by CCR. This provides a clinically relevant and actionable insight that can go beyond simple meal detection.

- The cross-validation strategy, while needing better justification in the main text, is a good setup for this low-data regime. Additionally, though there are other important contributing factors, such as sex and BMI,  the paper's idea to sort subjects by HbA1c and use a moving window for testing evaluates the model's ability to generalize to unseen subjects with similar metabolic profiles.

**Summary Of The Paper:**

This paper proposes a machine learning framework called RankPPGR to automatically estimate the macronutrient composition of a meal from its postprandial glucose response (PPGR), as measured by a continuous glucose monitor (CGM). As the authors have correctly alluded to, the high inter-individual variability in PPGRs makes it difficult to build models that generalize across subjects. To address this, the paper frames the problem as one of rank learning. The authors train a Siamese network on pairs of PPGRs from the same individual to learn which meal has a higher glycemic load, defined by the carb-caloric-ratio (CCR), with the aim to make this model personalized and robust to individual differences in metabolism etc. The framework also includes an aggregator network that uses these pairwise rankings against a set of reference meals to perform few-shot regression and predict the actual CCR value. Finally, the paper extends this model to learn a weighted CCR (WCCR), optimizing the contribution of protein, fat, and fiber to the glycemic load, rather than assuming a uniform caloric contribution. The authors evaluate their approach on CGMacros (with 45 subjects that have varying metabolic health status).

**Weaknesses:**

**Major Comments**:

- The paper claims to address "automatic diet monitoring," a very broad task. However, the experiments are confined to only breakfast shakes, which is noted in a footnote and not in the main text. The exclusion of this more challenging, free-living data weakens the paper's central claims. It is not clear if the model's performance would hold on meals with more complex compositions and preparation methods. The findings, while a good proof of concept, are currently only demonstrated in a narrow and highly-controlled setting.

- Another major weakness is the ambiguity of the few-shot regression methodology and its impact on the results (e.g. in Table II). While the aggregator shows a relative improvement over the baseline, its absolute performance is modest (R² of 0.25, RRMSE of 0.2). This indicates the model still struggles to make precise predictions. This modest performance makes the reference meal substitution strategy, described in footnote 2 (the PDF does not have page numbers to refer to) concerning. If the model frequently "borrows" reference PPGRs from phenotypically similar subjects in the training set, these already-modest results may be artificially inflated. Without a quantification of this effect, the validity of the paper's core regression claims is difficult to assess.

**Minor Comments**
- I think improving the presentation of results and figures can help convey the message of the paper better. For example, the flow of Figure 1 with (a), (b), (c), (d), and (e) is unintuitive.
- Fig 1's caption has a mistake: the figure uses C, P, B, and F to denote Carbs, Protein, Fiber and Fat, but the caption says "(Carbs C, protein P, fat F, and fiber F).
- Figure 1 seems to have some edges and shadows on the bottom of the plot, which I don't think are intended by the authors.
- It would be good to also report MAE in addition to the other regression metrics
- It would help the reviewers if the authors could add page numbers to the PDF.
- Figure 1 (d): both $g_1(t)$ and $g_2(t)$ lead to $p_1$ and $p_1$; $g_2(t)$ should lead to $p_2$.

**Really Minor Comments**
- There is an instance where the quotation marks are both `"`, which shows both quote and unquote as "backwards" unquote. You can fix this in latex by having ``fixed" (instead on "fixed").
- Please consider properly capitalizing words in references; e.g. IDF is shown as "idf", or "CGMacros" is shown as "Cgmacros". You can do this the bib file by adding `{}` around the word you want to capitalize.

---

### Official Review · Reviewer_JiTz · 2025-07-18
**Review for RankPPGR: automatic diet monitoring through rank learning**

**Confidence:** 4
**Clarity Of Writing:** good
**Clinical Significance:** good
**Methodological Novelty:** great
**Overall Rating:** 5

**Experiments And Results:**

good

**Questions For The Authors:**

How does model performance degrade when reference meals are limited or missing entirely? If the number of available reference meals per subject is reduced to, say, 2 instead of 5, how does that affect classification and regression performance? A response demonstrating robustness to limited reference data could increase confidence in real-world applicability and may positively impact the score.

Have the authors explored incorporating phenotype features (e.g., A1c, BMI, age) into the model directly (e.g., via FiLM layers), even in this initial study? Understanding whether this improves generalization across subjects would be useful. A demonstration of such an extension—even in a preliminary form—would strengthen the paper and may influence the overall score.

**Strengths:**

The paper addresses a clinically relevant problem of automating diet monitoring for diabetes management using CGMs. The rank-learning formulation is a compelling way to mitigate inter-individual variability, which is a major obstacle in modeling PPGRs. The architecture is well-motivated and the methodological components (e.g., pairwise training, few-shot aggregator, and learnable macronutrient embedding) are integrated cleanly. The analysis on the contribution of individual macronutrients to PPGRs is both interpretable and actionable from a nutrition science perspective. Results are clearly presented and statistically significant improvements over the baseline are demonstrated.

**Summary Of The Paper:**

This paper introduces RankPPGR, a rank-based learning framework for inferring the macronutrient composition of meals from postprandial glucose responses (PPGRs) captured via continuous glucose monitors (CGMs). The authors cast the inverse problem as a pairwise ranking task, rather than a direct regression problem. RankPPGR uses a Siamese neural network to compare PPGRs within subjects and learn embeddings that reflect the glycemic load of meals. An aggregator model then uses the pairwise rankings to perform few-shot inference of macronutrient composition. The paper further extends this approach by learning a weighted carb-caloric-ratio (WCCR) that captures non-uniform effects of protein, fat, and fiber on glycemic response. The model is evaluated on the CGMacros dataset of 45 participants consuming controlled meals, and is shown to outperform a direct regression baseline on both classification and regression metrics.

**Weaknesses:**

The primary limitation is that the evaluation, while thorough, is confined to a relatively small dataset (45 subjects) with controlled breakfast meals only. This raises concerns about generalizability to real-world, less constrained dietary settings. The current setup also requires phenotype-matched reference meals, which may limit applicability in fully automated or cold-start scenarios. While the model’s few-shot setup is a pragmatic compromise, further ablation on the required number and diversity of reference meals would help clarify practical deployment requirements. Additionally, while the paper justifies the use of within-subject comparisons to mitigate variability, it would be valuable to evaluate performance in out-of-subject inference with demographic or health covariates included (as suggested in the discussion). The interpretability of learned WCCR weights is a strength, but the clinical significance of differences (e.g., how much improvement WCCR offers over CCR in practice) could be elaborated.